# How Does Code Pretraining Affect Language Model Task Performance?

**Jackson Petty**[*]                                                           *research@jacksonpetty.org*
*Department of Linguistics*
*New York University*

**Sjoerd van Steenkiste**                                                    *svansteenkiste@google.com*
*Google Research*

**Tal Linzen**                                                                      *linzen@google.com*
*Google Research*

**Reviewed on OpenReview:** *https://openreview.net/forum?id=pxxmUKKgel*

## Abstract

Large language models are increasingly trained on corpora containing both natural language and non-linguistic data like source code. Aside from aiding programming-related tasks, anecdotal evidence suggests that including code in pretraining corpora may improve performance on other, unrelated tasks, yet to date no work has been able to establish a causal connection by controlling between language and code data. Here we do just this. We pretrain language models on datasets which interleave natural language and code in two different settings: *competitive*, in which the total volume of data seen during pretraining is held constant; and *additive*, in which the volume of language data is held constant. We study how the pretraining mixture affects performance on (a) compositionality, measured by generalization accuracy on semantic parsing and syntactic transformation tasks, and more broadly on (b) downstream non-code-related objectives, measured by performance on tasks from the BigBench benchmark. We find that pretraining on higher proportions of code improves performance on compositional tasks involving structured output (like semantic parsing), and mathematics. Conversely, increased code mixture can harm performance on other tasks, including on tasks that require sensitivity to linguistic structure such as syntax or morphology, and tasks measuring real-world knowledge.

## 1 Introduction

Large language models (LLMs) are increasingly used not only as natural-language assistants, but also for programming. LLMs which are trained on corpora containing code in various programming languages are used as programming assistants capable of generating code from natural-language descriptions (Chen et al., 2021), translating code between programming languages (Roziere et al., 2020), decompilation of machine code into human-readable source code (Hosseini & Dolan-Gavitt, 2022), repairing vulnerabilities in existing code (Pearce et al., 2023), and even acting as programming agents when paired with tools (Yang et al., 2024a). These use cases have motivated adding code to pretraining corpora (see, *inter alia*, Gemini Team et al. 2024; OpenAI et al. 2024; Anthropic AI 2024; Groeneveld et al. 2024).

Concomitant to the inclusion of code in pretraining corpora, the performance of LLMs on many tasks has improved. Relevant for our purposes, many of the best-performing models include code in their pretraining corpus (see, *inter alia*, Fu & Khot 2022; Ye & Durrett 2022; Ye et al. 2023; Zhang et al. 2023; Zhou et al. 2023; Kim et al. 2024; Ma et al. 2024; Yang et al. 2024b; Razeghi et al. 2024; Coda-Forno et al. 2024; Longpre

---
[*]Work done while a student researcher at Google.

et al. 2024). That models trained in part on code perform well on several non-programming benchmarks raises intriguing questions: Does pretraining on code confer an advantage on non-programming tasks? If so, given a fixed compute budget, how much data should be allocated to code instead of natural-language data?

Establishing a causal relationship between code pretraining and downstream performance is difficult. Earlier studies have tackled these questions by comparing off-the-shelf code and no-code models (see, *inter alia*, Kim et al. 2024; Coda-Forno et al. 2024). Such observational studies are limited by the design choices of model creators and the availability of information about hyperparameters and training data. Many of the models typically surveyed are proprietary and don't disclose this information. While pairs of open-source models differing only in their pretraining corpora do exist, such as Llama 2 & Code Llama (Touvron et al., 2023; Roziere et al., 2023) or Gemma & CodeGemma (Gemma Team et al., 2024; Google, 2024), they often come with two important caveats: first, the code-variants of the models are derived by taking the non-code variants and conducting additional pretraining on code data, meaning the comparisons cannot control for total data volume; second, each pair treats the inclusion of code data as a binary variable, either present or absent, frustrating attempts to explore how changes in the *amount* of code influence downstream behavior.

We address these issues directly. We construct datasets that mix natural-language and source-code data at varying ratios, treating code inclusion as a continuous variable. We then pretrain language models of equal size on these parameterized datasets in two different experimental setups: a *competitive* setting where we keep the total volume of training data constant and vary the percentage allocated between code and natural language; and an *additive* setting where we keep the volume of language data constant and add additional amounts of code on top.

Previous work has found that augmenting training data with synthetic formal languages instantiating compositional patterns can improve compositional generalization (Papadimitriou & Jurafsky, 2023; Yao & Koller, 2024; Lindemann et al., 2024). Like formal languages, source code has a number of qualities that may aid models on seemingly unrelated tasks: it is highly structured, by virtue of its conformance to the syntax of the programming language it's written in; it is generally high-quality, owing to the use of linting and bug-checking tools and programming methodologies employed by its authors; it has interpretable semantics which is grounded by the functionality it describes; and, notably for compositionality, it contains instances of identical arguments and functions (e.g., variable names and method signatures). Informed by these observations, we evaluate our trained models for compositional generalization by finetuning them on three compositional generalization benchmarks (COGS, COGS-vf, and English Passivization). We also measure their performance on a broad array of tasks from BigBench to see how well code helps or hurts performance on unrelated domains.

We find that including code in a model's pretraining corpus has noticeable impacts on its performance on downstream tasks, in varying directions. Higher code mixtures improve performance in arithmetic and compositionality in domains whose output has formal structure (like semantic parsing). Conversely, increased exposure to code can harm language model performance on purely-linguistic tasks and tasks involving factual knowledge. We conduct permutation tests to study the impact of pretraining on downstream tasks and show that code pretraining increases the variance on task performance while raising the performance on the upper-quartile of tasks.

## 2 Related Work

Earlier work has studied whether pretraining on code is beneficial for non-programming tasks. Observational studies have looked at the impact of code on downstream performance post-hoc. Fu & Khot (2022) speculated that code pretraining is at least partially responsible for the improvement in capabilities between the `-001` and `-002` series of GPT-3(.5) models, specifically highlighting chain-of-thought reasoning, long-term dependency sensitivity, and "complex reasoning" as likely resulting from code pretraining. Yang et al. (2024b) provides a broad study of how code impacts language model capabilities, arguing that code improves complex reasoning and structured data understanding. Mueller et al. (2024) shows that code pretraining improves generalization on syntax-sensitive in-context learning tasks. By contrast, Coda-Forno et al. (2024), in an observational study, conclude that code pretraining does *not* improve model performance on a benchmark of behavioral

tasks motivated by cognitive psychology. Kim et al. (2024) show that code pretraining improves models' entity-tracking capabilities.

Several experimental studies on the impact of code pretraining have also been conducted. Ma et al. (2024) attempt to verify the impact of code experimentally, comparing the 2.6 B parameter CodePanGu2.6 model trained on a mixture of natural-language and code data to Zeng et al. (2021)'s 2.6 B and 13 B parameter PanGu models of the same architecture trained only on natural language data. They conclude that code exposure, both during pretraining and instruction finetuning, is beneficial for performance on logical, legal, analogical, and scientific reasoning, and for chain-of-thought capabilities, though their experimental design does not control for data volume ($\sim$26.5 B tokens for PanGu2.6/13 versus[1] $\sim$42 B tokens for CodePanGu2.6) and does not quite control for model and training hyperparameters (models differ in the number of attention heads and use slightly different optimizer settings, which are magnified by the large difference in the number of training steps due to the difference in dataset size). Ma et al. (2024) also show exposing code to models early on during training can be helpful for some tasks. Longpre et al. (2024) show experimentally that removing code from a model's pretraining corpus harms performance on question answering in a number of different domains, though their experimental setup does not control for data volume and, consequently, other training hyperparameters sensitive to this.

The closest study to ours is the concurrent Aryabumi et al. (2024), which provides a thorough examination of the impact of mixed text-code training on reasoning and natural language tasks when controlling for data volume. We differ from Aryabumi et al. (2024) in three ways. First, in addition to comparing training setups where total data volume is fixed (our *competitive* case, and all training recipes studied in Aryabumi et al. 2024), we also consider training mixtures which vary the amount of code while keeping the volume of language data fixed (our *additive* case). Second, we treat dataset composition as a continuous range and conduct regression analysis to quantify the impact of code mixture. Third, we focus particularly on studying how code pretraining impacts compositional generalization, motivated by the hypothesis that code instantiates compositional patterns useful for generalization. We compare our results and those of Aryabumi et al. (2024) in Section 6.

## 3 Dataset Construction

To study how the amount of code in a language model's pretraining corpus impacts downstream performance, we construct datasets which interleave natural language and code sequences. The ingredients for our datasets are the English portion of the Colossal Cleaned Common Crawl (C4; Raffel et al. 2020) and a version of the code portion of The Pile (Gao et al., 2020), itself taken from public GitHub repositories; we use a version which has been cleaned to include only non-binary files smaller than 1MB with common code-related file extensions.

Each dataset, which we refer to as a 'code mixture,' is parameterized by a single value $m \in [0, 1]$ representing the percentage of code in the training data, under the assumption that the C4 dataset has been fully cleaned of any code data. The mixture $m$ relates the number of total tokens $N_{\text{total}}$ in the dataset to the number of code $N_{\text{code}}$ and language $N_{\text{lang}}$ tokens via

$$N_{\text{code}} = m \cdot N_{\text{total}}, \qquad N_{\text{lang}} = (1 - m) \cdot N_{\text{total}}.$$

We construct families of training datasets in two different settings: *competitive*, in which the total amount of data is held constant while $m$ varies, reducing the number of language tokens as the number of code tokens increases; and *additive*, in which the number of language tokens is held constant while the number of code tokens increases proportional to $m$ (see fig. 1).

---

[1] There is some ambiguity in the way Ma et al. (2024) describe their dataset: first, they cite that PanGu13 is trained on 1TB of data, but Zeng et al. (2021) report that it is trained on 100GB of data while their far larger 200 B parameter model is the one trained on 1TB of data; second, Ma et al. (2024) detail the individual data sources in GB but report the total dataset size in terms of tokens. It is unclear from phrasing whether their sampling strategy yields a dataset of 100 GB *in total*, or contains 100 GB of text data in addition to 50 GB of code data, but in either case the Table 4 in Zeng et al. (2021) shows that the 100 GB natural-language dataset used for the PanGu comparison models contains only $\sim$26.5 B tokens, compared to CodePanGu's $\sim$42 B tokens.

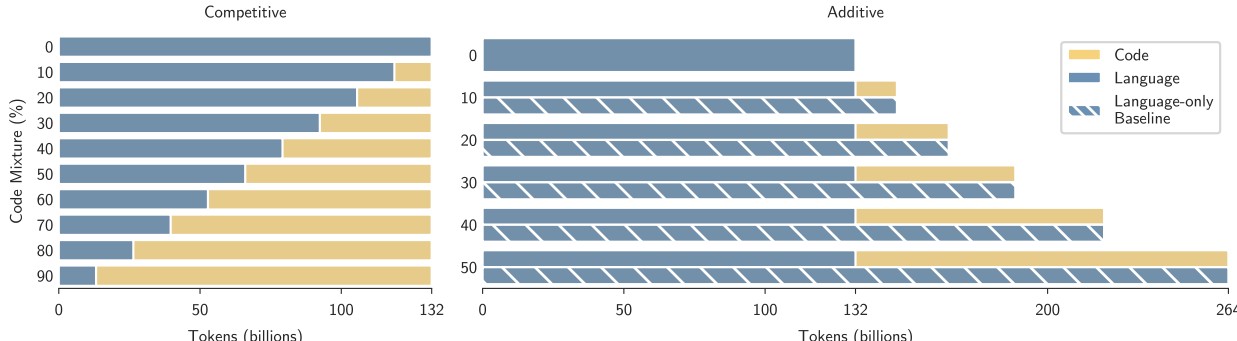

Figure 1: Code mixtures for the competitive and additive settings; for comparability between settings, we set the 0% mixture in both settings to have 132 B tokens of natural-language data (so $N_{\text{total}} = 132$ B tokens in the competitive setting, and $N_{\text{lang}} = 132$ B tokens in the additive setting). Note that these sequences are shuffled during training, so models see code and language data at the same time.

**Competitive:** Here, $N_{\text{total}}$ is held constant while $m$ varies between 0 and 0.9. This means that models trained on the 0% code mixture see $N_{\text{total}} = N_{\text{lang}}$ language tokens and 0 code tokens, while those trained on the 90% mixture see $0.1 \times N_{\text{total}}$ tokens of language data and $0.9 \times N_{\text{total}}$ tokens of code data.

This setting provides the clearest way to quantify the marginal utility of training on code instead of language, since we control for the total volume of data seen and consequently the total compute cost. However, the interpretability of results on mixtures with high values of $m$ may be diminished since removing nearly all natural-language training data from a model's training corpus will lessen its ability to interpret and generate language; this, in turn, may greatly reduce its utility, even on code-related tasks, since the model will have far less ability to understand prompts or follow instructions. Additionally, the applicability of any results here to established pretraining setups may be limited by the fact that it will always be better in an absolute sense (and may be in a compute-optimal sense) to train a model on more data rather than less data (see, for instance, the conclusions of Hoffmann et al. 2022). Given this incentive, artificially limiting the amount of either code or language data provided to a model may not accurately reflect the considerations of model developers who, if they want to improve the code performance of a model, will simply add additional code data to the training corpus. To mitigate these issues, we also consider a second setting:

**Additive:** Here, $N_{\text{lang}}$ is held constant while $m$ varies between 0% and 50%. In order to keep $N_{\text{lang}}$ fixed while $m$ varies, we increase the number of total tokens proportionally:

$$N_{\text{total}} = N_{\text{lang}} \times \frac{1}{1-m}.$$

Since $N_{\text{total}}$ increases unboundedly in $m$, we limit our study to consider additive mixtures of at most 50% code, which have twice as many tokens as the 0% mixture, which is identical to the 0% competitive mixture. This setting guarantees that all models have seen the same amount of natural language data, ameliorating the concern that any degradation in performance may result from insufficient exposure to natural language, but at the cost of failing to control for total data volume or compute. To further ensure that we can adequately compare code and non-code models across, we construct language-only baseline datasets for each code mixture. These datasets have the same number of total tokens, but with 100% of those tokens coming from natural language.

Though both the code and language datasets we use are intended to be distinct in content type from one another, it is likely that there is a degree of overlap in content type between them. For instance, source code often contains comments and string literals containing natural-language data; in the other direction, though C4 is cleaned using a variety of heuristics, some of which are explicitly designed to exclude code-like data, it is likely that such cleaning attempts are imperfect and therefore there may be a (relatively) small amount of code data in the natural-language data source. We do not perform any additional filtering or cleaning of these

data sources. As such, the code data source almost certainly contains some natural language data, just as the natural language source likely contains some un-filtered source code. We leave it to future work to explore what impact, if any, fully removing code comments has on downstream performance, though we speculate that the pairing of a code block with a natural language description of its function and implementation may be meaningfully important to language model performance on non-code tasks whose performance is aided by code pretraining.

## 4 Experimental Setup

### 4.1 Model Construction & Training

We use the datasets constructed in section 3 as pretraining corpora for causally-masked decoder-only transformer language models (Vaswani et al., 2017; Radford et al., 2019). We construct 12-layer decoder-only models in `t5x` Roberts et al. (2023). Model hyperparameters were chosen following the methodology of Wang et al. (2022) and Petty et al. (2024) to approximate decoder-only versions of T5-large, resulting in models with roughly 374 M parameters; see Appendix A for hyperparameter details.[2] We pretrain these models with a base natural language data volume of 132 B tokens. This means that all models in the competitive setting were trained with $N_{\text{total}} = 132$ B tokens, while the models in the additive setting were trained with $N_{\text{lang}} = 132$ B tokens, and hence $N_{\text{total}}$ varying between 132 B tokens and 264 B tokens depending on the mixture; we use a batch size of 128, meaning that models were trained for between 1 M and 2 M steps, depending on the mixture and setting. For each combination of code mixture and setting, we pretrain models from five different random seeds. We pretrain models on TPUs. We estimate that full replication of the pretraining procedure outlined here would take roughly 750 TPU-days of compute.

### 4.2 Evaluation

We measure performance on three compositional generalization benchmarks and, more generally, on Big-Bench tasks. For each evaluation domain, we quantify the impact that code pretraining has on performance by calculating lines of best fit between performance (e.g., generalization accuracy for the compositional generalization benchmarks or multiple-choice grade for BigBench multiple choice tasks) and code mixture.

#### 4.2.1 Compositional Generalization

Compositional generalization is a measure of how well a learner can generate and interpret novel, licit combinations of primitive pieces which have been previously learned. Originally motivated to describe human linguistic faculty—such as the ability of speakers to produce and understand an infinite number of novel, grammatical sentences—compositionality is also a relevant property of many formal systems, like mathematics or programming languages. We hypothesize that the presence of source code in pretraining data may aid models in making this kind of generalization since source code often contains sequences in which a finite set of primitives (e.g., variable and method identifiers) are broadly combined.

To evaluate whether increased code mixture enables compositional generalization, we finetune our pretrained models on a suite of compositional generalization datasets: COGS (Kim & Linzen, 2020), a semantic parsing task in which natural-language sentences are transformed into a formal semantic representation; COGS-vf (Qiu et al., 2022), a variant of COGS which simplifies the output format; and English Passivization (Mueller et al., 2022), a natural-language transduction task in which synthetically-generated active-voice sentences are transformed into passive variants. Each dataset contains training, validation, and generalization splits, where the generalization split is constructed to test licit-but-unattested combinations of familiar primitives. Table 1 shows examples of the input and output sequences for each of the datasets.

COGS and COGS-vf both divide their generalization split into two parts based on generalization type: either *lexical*, in which a known primitive is used in a grammatical position it has not been seen in before (e.g.,

---

[2]We don't view our paper as exploring the merits of the specific architecture in question; our results here are of interest irrespective of the specific transformer model architecture, unless there is a reason to suspect that a change in architecture confers an inductive bias which is particularly sensitive to source code as a training distribution.

| COGS | $x$: A hedgehog ate the cake . |
| | $y$: $^*$cake$(x_4)$; hedgehog$(x_1)$ AND eat.agent$(x_2, x_1)$ AND eat.theme$(x_2, x_4)$ |
| COGS-vf | $x$: A hedgehog ate the cake on the bed . |
| | $y$: eat(agent = hedgehog, theme = $^*$cake(nmod.on = $^*$bed)) |
| English Passivization | $x$: our vultures admired her walrus above some zebra . |
| | $y$: her walrus above some zebra was admired by our vultures . |

Table 1: Examples of inputs ($x$) and targets ($y$) from each compositional generalization dataset.

*hedgehog* in subject position, when it had only been seen during training as an object); or *structural*, in which a known grammatical structure is used in a novel position (e.g., a prepositional phrase such as *on the mat* modifying the subject, when in training such phrases only modified objects). Previous studies involving COGS and COGS-vf have found the structural generalization examples in COGS to be much harder than the lexical generalization examples. Reducing the complexity of the output form, as is done in COGS-vf, makes the structural tasks somewhat *easier*, though not *easy*. Petty et al. (2024) found that models of a comparable size could attain accuracies near 90% on the lexical generalization examples from COGS but near 0% on the structural examples; on COGS-vf, models were able to attain accuracies greater than 95% on lexical cases and 10% on structural cases.

For all compositional generalization datasets, we finetune models for 10 K steps and report the mean full-sequence accuracy (i.e., 1 if every autoregressively-generated token is correct, 0 otherwise) over all examples in the generalization split for each random pretraining seed. In general, we do not observe any meaningful effect that code mixture has on training dynamics over the course of finetuning, nor do we see evidence of over- or under-fitting to the validation or generalization sets; see Appendix C for a full discussion.

### 4.2.2 BigBench

We also evaluate models on BigBench (Srivastava et al., 2023), a benchmark of 204 diverse and challenging tasks presented in a common format. We evaluate models in a zero-shot setting, where a question is given in context (e.g., `What is 697 times 205?` from the *3-digit multiplication* task) and the model must either generate the correct label (e.g, `(a).`) from a provided list of responses (for multiple-choice tasks) or generate the correct answer (for generative tasks). Since our focus is on the effect of code in pretraining on non-code tasks, we exclude from consideration tasks which are explicitly designed to test the capabilities of models at understanding or generating source code. Table 2 shows examples of the input and output sequences for the BigBench tasks we discuss in detail.

## 5 Results

**Code improves compositional generalization for structured outputs.** When we finetune on COGS and COGS-vf, where the output domain has a formal structure, we find that performance improves as the proportion of code increases in both the competitive and additive settings (see fig. 2 and table 3 in Appendix B). The effect is most pronounced for the structural generalization examples from COGS-vf in the competitive and additive settings (regression coefficients $\hat{\beta} = 0.147$ and $\hat{\beta} = 0.165$, respectively; this indicates that the best-fit line predicts an accuracy increase of 14.7% as the proportion of code increases from 0% to 100%), though all code-mixture models show a non-negative relationship between code mixture and generalization accuracy. Code helped the least on the structural generalization examples from COGS, where absolute performance remained near-zero. In the additive setting, we find that code-mixture models perform as well (on lexical generalization examples) or better (on structural generalization examples) than the equivalent language-only baseline models.

In order for models to generalize compositionally, two things must happen: first, models must correctly generalize the distribution of arguments and predicates to match the true-but-unseen patterns of composition (e.g., they must learn that syntactic objects become arguments to 'theme' for all primitives, even those only previously seen as subjects); and they must produce well-formed outputs. Kim & Linzen (2020, §G.2) note

| `bb-arithmetic` | $x$: What is 68824 times 42716? |
| | $y$: 9033448237, 3839424324, 18962582, 564059290599, banana, house, **2939885984** |
| `bb-common-morpheme` | $x$: What is the common morpheme among these words: pyre, empyrean, antipyretic, pyrotechnics? |
| | $y$: **fire**, hot, oxygen, medicine |
| `bb-fantasy-reasoning` | $x$: Long ago you had sold your soul to the devil, but the postal service was so utterly bad that they had lost the package where your soul was. Since the transaction was completed before it, you have the benefits of the deal while the devil still has no control over you. Does the devil have any control over your soul now? |
| | $y$: Yes, **No** |
| `bb-general-knowledge` | $x$: How many legs do horses have? |
| | $y$: two, **four**, six, three, one, none |
| `bb-implicatures` | $x$: Does Speaker 2's answer mean yes or no? Speaker 1: 'But aren't you afraid?' Speaker 2: 'Ma'am, sharks never attack anybody.' |
| | $y$: yes, **no** |

Table 2: Examples of inputs ($x$) and answers ($y$) from selected multiple-choice BigBench tasks. Correct answers are bolded.

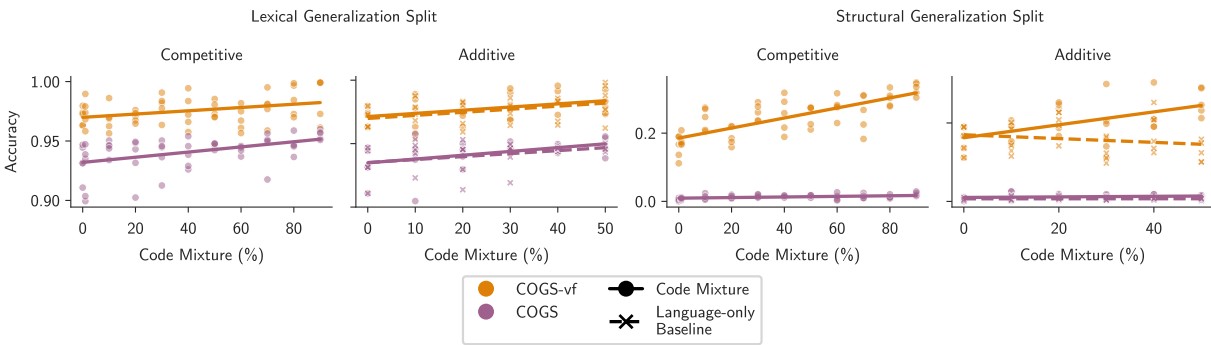

Figure 2: Full-sequence accuracy on the generalization set increases with code mixture on COGS and COGS-vf in both the competitive and additive settings. In the additive setting, code-mixture models outperform language-only baselines on the harder structural generalization cases. In all cases, validation accuracy is 100%.

that Transformer models in particular often failed at producing syntactically well-formed logical expressions for the generalization examples in COGS. Since code has similar syntactic requirements to those of COGS logical expression (e.g., well-balanced parentheses), the improvement we observe in generalization accuracy may be due to improvements in the well-formedness of outputs, rather than due to better compositional generalization. To test this hypothesis, we compute a very high-level measure of syntactic well-formedness for model outputs—namely, whether or not the decoded logical forms have well-balanced parentheses—and examine how well-formedness varies by code mixture.

Figure 3 shows that exposure to code does not, in general, improve the well-formedness of generalization outputs. Only on structural generalization examples from COGS-vf in the additive setting does the regression coefficient $\hat{\beta} = 0.049$ exceed 0.01; for all other code-mixture models, increased code mixture has a near-zero or negative impact on syntactic well-formedness (table 4). This means that the observed relationship between higher code mixture and generalization accuracy is attributable to models learning better generalizations for argument distribution rather than merely producing more well-formed outputs.

**Code improves performance on arithmetic, up to a point.** On multiple-choice multi-digit arithmetic tasks from BigBench, increased code mixture generally has a positive impact on performance. In both

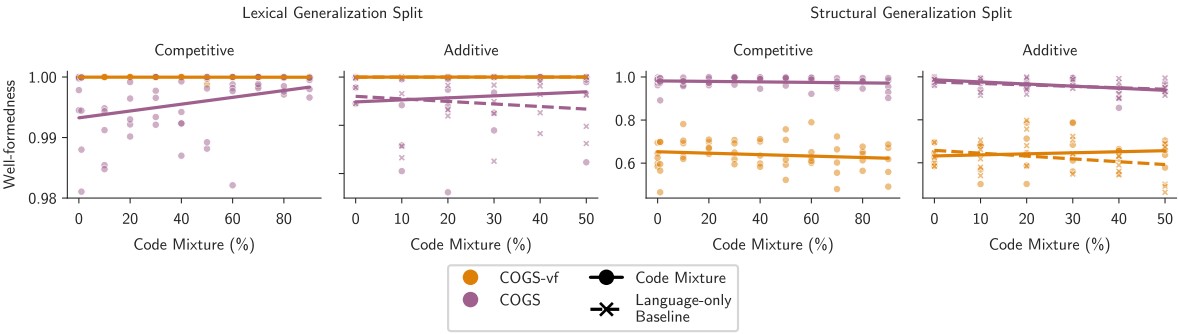

Figure 3: Pretraining code mixture has little impact on the well-formedness of generalization outputs in any setting.

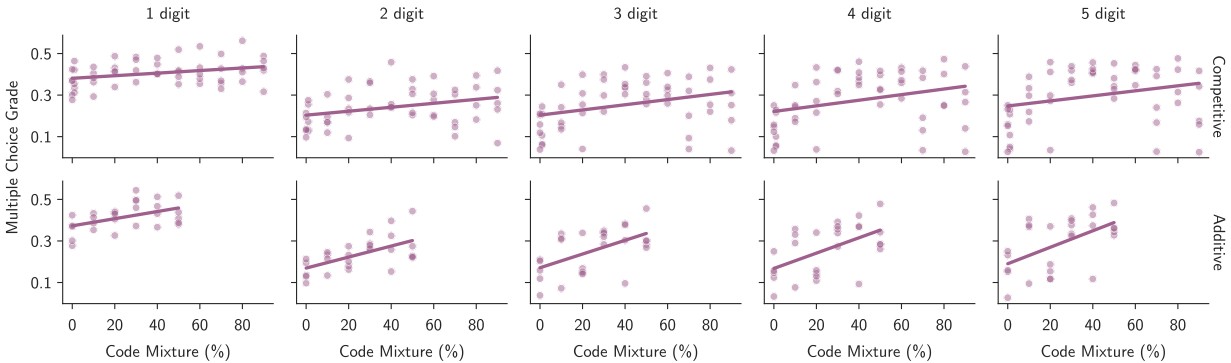

Figure 4: On multi-digit multiple choice arithmetic tasks, performance modestly increases with code mixture in the additive setting, while it increases then decreases in competitive. In both settings, the effect is more pronounced as the number of digits (rows) increases.

competitive and additive settings, higher code mixture results in greater multiple-choice accuracy, with the impact growing more pronounced as the number of digits increases (see fig. 4 and table 6). In the competitive setting, performance peaks at a code mixture between 40% and 50% and thereafter tends to decrease, though the overall trend remains positive; this inverted-U shaped performance curve also grows more pronounced as the number of digits increases.

**Code distracts from linguistic- and world-knowledge.** We also identify cases where increased exposure to code *harms* performance by looking for tasks whose performance is negatively correlated with code mixture. These tasks include ones which involve purely linguistic knowledge (such as the English Passivization compositional generalization task as well as the Implicatures and Common Morpheme BigBench tasks) as well as those which involve reasoning or world-knowledge (such as the General Knowledge and Fantasy Reasoning BigBench tasks).

Figure 5 shows this negative trend on the English Passivization compositional generalization benchmark, where performance (as measured by mean full-sequence accuracy on the generalization split) decreases as code mixture increases in both the competitive and additive settings. Furthermore, in the additive setting the language-only baseline models outperform the code-mixture models. See table 5 for exact regression coefficients.

These negative trends show that increased exposure to code during pretraining does not uniformly improve the ability of language models to generalize compositionally independent of the output domain; whereas

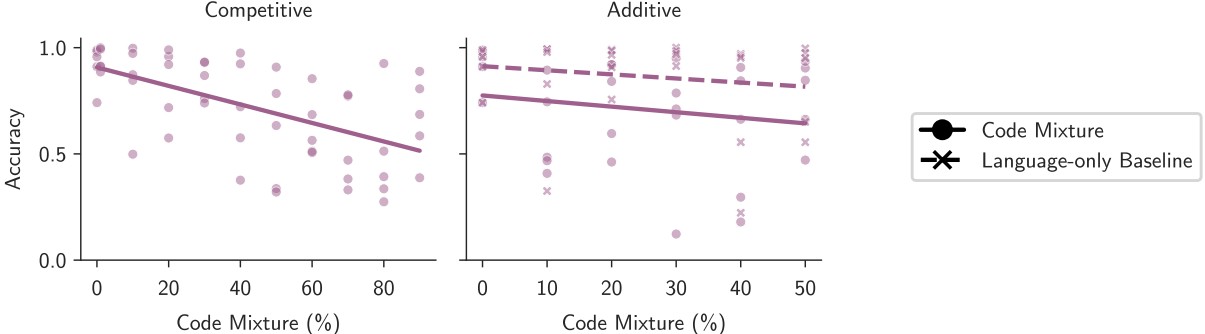

Figure 5: On English Passivization, a compositional generalization benchmark where (unlike COGS) both the inputs and outputs are in natural language, increased code mixture results in lower full-sequence generalization accuracy in both settings. In the additive setting, code-mixture models underperform language-only baselines on the harder structural generalization cases.

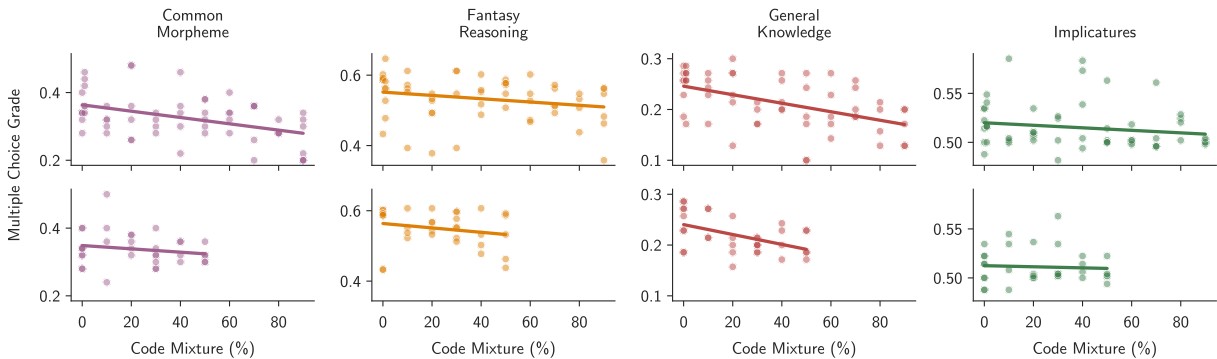

Figure 6: On a variety of BigBench tasks involving linguistic or factual knowledge, increased code mixture reduces accuracy.

COGS and COGS-vf, whose output domain is formal logic expressions, benefit from increased code exposure, generalization tasks which involve natural-language output domains appear to obviate any compositionality benefit conferred to models through code exposure. This may make intuitive sense, as decreased exposure to natural language data (in either an absolute or relative sense) may reduce any linguistically-relevant inductive biases models need, in partial conflict with Mueller et al. (2024)'s finding that code pretraining aids syntax-sensitive generalization for *in-context learning* tasks.

We also find instances of BigBench tasks where code mixture is negatively correlated with performance; Figure 6 highlights four such tasks where increased exposure to code during pretraining harms performance in both competitive and additive settings. See table 7 for exact regression coefficients.

## 5.1 The impact of code in aggregate

The results presented above highlight particular cases where code mixture has a noticeable impact on performance, but how does code pretraining affect the remaining BigBench tasks? We want to know how code pretraining impacts performance in aggregate for two reasons. First, we want to know if adding code helps *in general*: is adding code helpful or harmful for most tasks? Second, since it's likely that following any type of intervention models will be better at some tasks and worse at others than before the intervention, we want to confirm if the effects of code we observe are statistically significant or could have arisen due to chance.

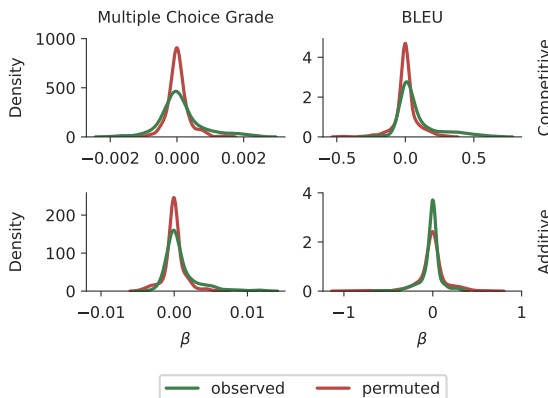

Figure 7: Kernel Density Estimates for the slopes $\beta$ of linear regressions between task performance and code mixture on BigBench tasks.

To answer this, we perform a permutation test on the slopes derived above from best-linear-fits of task performance versus code mixture. We start by taking the underlying performance-by-mixture data and shuffling the independent variable (code mixture) within each task and recompute slopes for the lines-of-best-fit. Figure 7 shows the distribution of slopes for the observed (treatment) and counterfactual, permuted (control) data for both settings and metrics. For multiple choice tasks in both settings and for generative tasks in the competitive setting, the distribution of treatment slopes (i.e., those observed) is less concentrated around 0 than the control distribution.

To quantify the difference between these distributions, we compute several different test statistics: the difference of means ($\Delta\mu$) as a measure of whether training on code improves task performance on average; the difference of variance ($\Delta$Var) as a measure of whether training on code increases the variance of task performance; the difference of skew ($\Delta$Skew) as a measure of whether training on code moves the distribution of task performance asymmetrically; and the differences in upper and lower quartiles ($\Delta$Upper/LowerQuartile) as a measure of whether training on code increases the model's performance on its best and worst-performing tasks.

We then perform two-sided permutation tests against the null hypothesis that the treatment and control distributions are drawn from the same underlying distribution by combining and randomly-repartitioning the samples 10 K times and recomputing each test statistic. We do this test independently for each setting (competitive and additive) and each BigBench question type: multiple choice (MCG) and generative (where performance is measured by BLEU).

Figure 8 shows the null distributions for each of the test statistics and the observed values for the multiple-choice questions in the competitive setting, along with the significance scores ($p$-values) for each statistic. We find a statistically significant difference of variance ($p = 0.0002$) and upper-quartiles ($p = 0.006$) at a significance level of $\alpha = 0.05$, indicating that increased code exposure in pretraining does have strong benefits for some tasks, while it increases the variance in downstream task performance in general. Other statistics measured were not significant at this significance level. Results are similar, in general, for other conditions.

## 6 Discussion

We find that including code in a model's pretraining corpus influences its performance on downstream, non-code tasks. Adding code improves performance on compositional generalization tasks whose output domain is highly structured, akin to the syntactic constraints of source code. Exposure to code during pretraining also improves performance on arithmetic tasks, an trend which grows more pronounced as the number of digits of the numbers included in those arithmetic tasks increases. Conversely, we also find tasks where increased exposure to code harms model performance, such as compositional generalization tasks involving

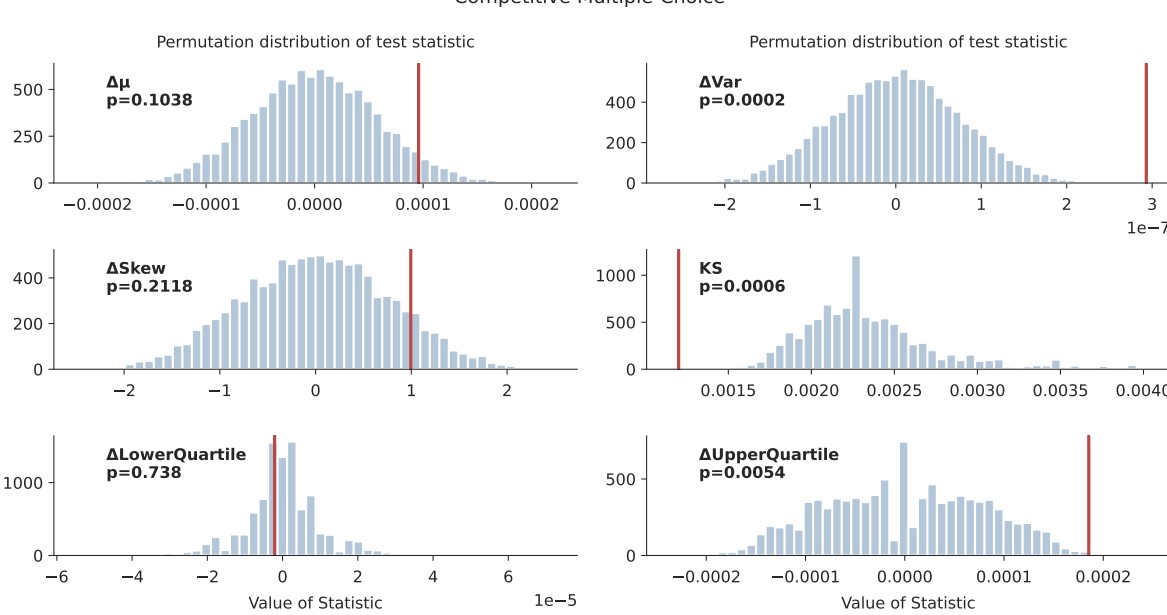

Figure 8: Null distributions (blue histogram) and observed values (red vertical rules) for various test statistics under a permutation test for slopes of performance by code mixture on Big Bench tasks with multiple choice grades in the competitive setting.

natural-language output or tasks involving linguistic or real-world knowledge. These trends appear in both a competitive setting, where increases in code data result in reduction of language data, and in a additive setting, where all models see a fixed amount of language data. This result is consistent with the observed degradation in performance increasing code mixture has on world-knowledge tasks in Aryabumi et al. (2024), and partially accords with their finding that performance on natural language tasks lessens for models trained on more than 25% code.

Despite the fact that code improves compositional generalization only in cases where the output domain is 'code-like,' we find that increased code exposure does not meaningfully improve the syntactic well-formedness of outputs in these cases; rather, the benefit conferred by code is to allow models to better learn the correct generalization for the distribution of arguments. We hypothesize that the deleterious impact of code on tasks involving linguistic or real-world knowledge comes from a reduction in linguistically-relevant inductive biases as models see less natural language data (either in an absolute sense in the competitive setting or a relative sense in the additive setting).

We conduct permutation tests on the distributions of per-task trend lines of performance-by-code-mixture to quantify the impact that code has on performance. We find that, in aggregate, training on code tends to improve performance on BigBench tasks at a statistically-significant level.

In light of these results, we suspect that the ideal combination of data sources depends on the intended domain of use for a model. Though our results show that adding in code to pretraining mixtures is on balance helpful, we do not weight any of the downstream evaluations by perceived importance, which may vary depending on the goals of model design. As a concrete example, a chatbot-like model should, according to our results, be trained on more code than is present in popular pretraining corpora like The Pile (Gao et al., 2020) or Dolma (Soldaini et al., 2024). This recommendation, however, should be caveated with the note that we do not explore how post-training objectives like instruction tuning (Wei et al., 2022) or RLHF (Ouyang et al., 2022) interact with code mixture in pretraining, which is a promising direction for future work.

### 6.1 Limitations

**Scale**   We survey relatively small models (374 M parameters), which limits our ability to establish how code pretraining affects capabilities which require models at the multi-billion parameter scale, like instruction following and advanced in-context learning. Aryabumi et al. (2024) additionally study models at a slightly larger scale (~2.8 B parameters) and observe similar conclusions. We also only consider pretraining corpora of between 132 B and 264 B tokens.

**Data Sources**   We treat 'code' and 'language' as a monolithic and disjoint data sources, but in reality source code contains linguistic data in the form of comments while natural language datasets may contain code-like structures even after cleaning and curation. It is possible that effect sizes would be increased with a more thorough separation of code and language data.

**Task Limitations**   We study a small set of tasks and evaluation modalities (fine-tuning on compositional generalization benchmarks and zero-shot performance on assorted BigBench tasks). Code pretraining may have impacts on other tasks, and those impacts may differ between fine-tuning, zero-shot, and multi-shot in-context learning.

### 6.2 Acknowledgements

We thank the anonymous reviewers for their helpful comments on previous versions of this paper.

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

## A Model Hyperparameters

We use the baseline $374\,\mathrm{M}$-parameter model configuration from Petty et al. (2024) for our experiments, which has $n_{\mathrm{layers}} = 24$, $d_{\mathrm{ff}} = 2816$, $d_{\mathrm{model}} = d_{\mathrm{attention}} = 1024$, and $n_{\mathrm{heads}} = 64$.

## B Regression Coefficients

| Dataset | Gen. type | Setting | Baseline | $\hat{\beta}$ | $\hat{\alpha}$ | $R^2$ |
|---------|-----------|---------|----------|---------------|----------------|-------|
| COGS | Lexical | Competitive | — | 0.022 | 0.932 | 0.776 |
| COGS | Lexical | Additive | False | 0.030 | 0.935 | 0.792 |
| COGS | Lexical | Additive | True | 0.024 | 0.935 | 0.869 |
| COGS | Structural | Competitive | — | 0.009 | 0.009 | 0.816 |
| COGS | Structural | Additive | False | 0.007 | 0.010 | 0.960 |
| COGS | Structural | Additive | True | 0.000 | 0.007 | 1.000 |
| COGS-vf | Lexical | Competitive | — | 0.014 | 0.970 | 0.877 |
| COGS-vf | Lexical | Additive | False | 0.025 | 0.971 | 0.816 |
| COGS-vf | Lexical | Additive | True | 0.024 | 0.970 | 0.851 |
| COGS-vf | Structural | Competitive | — | 0.147 | 0.186 | 0.413 |
| COGS-vf | Structural | Additive | False | 0.165 | 0.162 | 0.692 |
| COGS-vf | Structural | Additive | True | $-0.048$ | 0.170 | 0.961 |

Table 3: Coefficients of linear regressions $\hat{y} = \hat{\beta}x + \hat{\alpha}$ predicting generalization accuracy by code mixture on COGS and COGS-vf.

## C Additional Results

### C.1 Code does not help models learn faster

We chose to finetune models on the compositional generalization datasets for a fixed duration ($10\,\mathrm{K}$ steps) for simplicity of experimental design and to better facilitate comparison to earlier work which examined models on the same datasets Petty et al. (2024). Additionally, we store intermediate model outputs and checkpoints every $1\,\mathrm{K}$ steps to study if code-pretraining meaningfully changes the learning dynamics of models over the course of fine-tuning. We find that this is not the case: validation performance saturates quite early and does not diverge; generalization accuracy likewise reaches roughly its final value early and does not meaningfully change over the course of fine-tuning. We conclude that code-pretraining does not help a model reach higher performance on the compositional generalization benchmarks faster than it otherwise would independent from the effect code pretraining has on the model's final performance.

| Dataset | Gen. type | Setting | Baseline | $\hat{\beta}$ | $\hat{\alpha}$ | $R^2$ |
|---|---|---|---|---|---|---|
| COGS | Lexical | Competitive | — | 0.006 | 0.993 | 0.883 |
| COGS | Lexical | Additive | False | 0.004 | 0.995 | 0.988 |
| COGS | Lexical | Additive | True | $-0.005$ | 0.996 | 0.969 |
| COGS | Structural | Competitive | — | $-0.012$ | 0.982 | 0.980 |
| COGS | Structural | Additive | False | $-0.098$ | 0.986 | 0.718 |
| COGS | Structural | Additive | True | $-0.067$ | 0.976 | 0.860 |
| COGS-vf | Lexical | Competitive | — | 0.000 | 1.000 | 0.999 |
| COGS-vf | Lexical | Additive | False | 0.000 | 1.000 | 0.948 |
| COGS-vf | Lexical | Additive | True | 0.024 | 1.000 | 0.847 |
| COGS-vf | Structural | Competitive | — | $-0.033$ | 0.653 | 0.978 |
| COGS-vf | Structural | Additive | False | 0.049 | 0.632 | 0.987 |
| COGS-vf | Structural | Additive | True | $-0.132$ | 0.658 | 0.914 |

Table 4: Coefficients of linear regressions $\hat{y} = \hat{\beta}x + \hat{\alpha}$ predicting generalization well-formedness by code mixture on COGS and COGS-vf.

| Dataset | Setting | Baseline | $\hat{\beta}$ | $\hat{\alpha}$ | $R^2$ |
|---|---|---|---|---|---|
| English Passivization | Competitive | — | $-0.416$ | 0.894 | 0.718 |
| English Passivization | Additive | False | $-0.263$ | 0.775 | 0.966 |
| English Passivization | Additive | True | $-0.193$ | 0.913 | 0.973 |

Table 5: Coefficients of linear regressions $\hat{y} = \hat{\beta}x + \hat{\alpha}$ predicting generalization accuracy by code mixture on English Passivization.

| Dataset | # of Digits | Setting | $\hat{\beta}$ | $\hat{\alpha}$ | $R^2$ |
|---|---|---|---|---|---|
| BB Arithmetic JSON | 1 | Competitive | 0.062 | 0.381 | 0.906 |
| BB Arithmetic JSON | 1 | Additive | 0.172 | 0.373 | 0.780 |
| BB Arithmetic JSON | 2 | Competitive | 0.095 | 0.203 | 0.901 |
| BB Arithmetic JSON | 2 | Additive | 0.265 | 0.169 | 0.654 |
| BB Arithmetic JSON | 3 | Competitive | 0.124 | 0.204 | 0.891 |
| BB Arithmetic JSON | 3 | Additive | 0.330 | 0.171 | 0.706 |
| BB Arithmetic JSON | 4 | Competitive | 0.135 | 0.221 | 0.898 |
| BB Arithmetic JSON | 4 | Additive | 0.369 | 0.168 | 0.710 |
| BB Arithmetic JSON | 5 | Competitive | 0.121 | 0.248 | 0.925 |
| BB Arithmetic JSON | 5 | Additive | 0.397 | 0.190 | 0.706 |

Table 6: Coefficients of linear regressions $\hat{y} = \hat{\beta}x + \hat{\alpha}$ predicting generalization accuracy by code mixture on BB Arithmetic JSON.

| Dataset | Setting | $\hat{\beta}$ | $\hat{\alpha}$ | $R^2$ |
|---|---|---|---|---|
| BB Common Morpheme JSON | Competitive | $-0.093$ | 0.364 | 0.804 |
| BB Common Morpheme JSON | Additive | $-0.049$ | 0.349 | 0.968 |
| BB Fantasy Reasoning JSON | Competitive | $-0.047$ | 0.552 | 0.946 |
| BB Fantasy Reasoning JSON | Additive | $-0.062$ | 0.564 | 0.955 |
| BB General Knowledge JSON | Competitive | $-0.084$ | 0.246 | 0.749 |
| BB General Knowledge JSON | Additive | $-0.097$ | 0.240 | 0.759 |
| BB Implicatures JSON | Competitive | $-0.013$ | 0.520 | 0.971 |
| BB Implicatures JSON | Additive | 0.006 | 0.512 | 0.997 |

Table 7: Coefficients of linear regressions $\hat{y} = \hat{\beta}x + \hat{\alpha}$ predicting generalization accuracy by code mixture on BB Common Morpheme, Fantasy Reasoning, General Knowledge, and Implicatures JSON.

