# OpenReview forum: "How Does Code Pretraining Affect Language Model Task Performance?"
_TMLR — Accepted by TMLR_

### Review · Reviewer_DwKu · 2024-10-25

**Summary Of Contributions:**

This work studies the impact code data has (in pretraining datasets) on downstream task performance for LLMs. The amount of code in the pretraining corpus of LLMs is meticulously controlled with two different setups: a _competitive_ setting where the amount of pretraining data is kept constant (adding more code data reduces the amount of natural language data); and an _additive_ setting where the amount of natural language is kept constant (adding more code data increases the amount of pretraining data).
The resulting pretrained models are then finetuned on compositional generalization tasks (semantic parsing COGS), and more broadly on BigBench tasks. Results show that increasing code data during pretraining increases performance on structured output tasks (such as semantic parsing) and mathematical tasks, but decreases performance on more linguistic tasks and tasks involving real-world knowledge.

**Audience:**

Yes

**Broader Impact Concerns:**

When using code repositories as part of training data, one must be careful to use open and permissive license code. No information is provided about how the code was collected.

**Claims And Evidence:**

Yes

**Requested Changes:**

Nothing critical, below are suggestions to make the paper stronger.

**Strong Suggestion**: Give more information on the code dataset used. The mentioned ``_cleaned code from GitHub_’’ does not provide enough information for reproducibility purposes.

**Normal Suggestion**: Run a finetuning time analysis to see if pretraining on code helps finetuning models faster. Even if models reach similar final performance, faster training is often a desideratum.

**Weak Suggestion**: replace all the dots in Figures 2-6 with error bars, to have one dot per line per x-tick, and a vertical line showing the variance.

**Weak Suggestion**: Code is an interesting modality because it contains natural language in comments or text boxes in Python notebooks, thus helping LLMs bridge the gap between code and natural language instructions. Models trained only on Code have been known to understand natural language to some extent (see CodeGen and StarCoder projects). This could be mentioned in the related work sections of the paper.

**Question**:
Do you control for this (natural language present in code data) in your experiments? ie: did you remove all comments from the code you used during pre-training? – this can be answered in a paragraph detailing the code data collection used for pretraining.

**Strengths And Weaknesses:**

**Strenghts**

The paper is well-written and easy to follow.
This work is rigorous in its experimental settings, answering the research question of `How Does Code Pretraining Affect Language Model Task Performance?’
This work also provides a detailed statistical analysis showing that increased code exposure has, in general, performance benefits, but also increases performance variance across tasks.
The claims of this work are well supported by experimental evidence.

**Weaknesses**

1: The impact of this work is mitigated by the fact that its conclusion is similar to many previous works. As described in the related work section, many studies have already shown that including _some_ code data during pretraining is beneficial for many logical and reasoning downstream tasks. Despite controlling rigorously the amount of code during pre-training, this work has a similar conclusion: code improves performance on structured output tasks, but decreases performance on more linguistic tasks.

2: Experimental results seem very sensitive to the random initialization of the model weights. For instance in Figure 2, in the lexical generalization split of COGS (-vf), the competitive setting seems to suggest that a higher code ratio is beneficial, however in the additive setting we see that increasing the code ratio does as well as increasing the amount of natural language. The combination of these results is somewhat challenging to understand.

3: This work inspects the influence of pretraining dataset composition on the _performance_ of downstream tasks, but it could also influence the finetuning _time_ required to reach a certain performance. Some finetuning time analysis on downstream tasks could benefit this paper.

Below are some limitations of the work (all already acknowledged in the “limitations” section):

This work shows the benefits of code data during pretraining on only two very specific cases: structured output semantic parsing and multiplication tasks (to some extent). The work could somewhat benefit from additional experiments on multi-step reasoning or logical reasoning tasks as these are likely to benefit from code pre-training.

Despite trying to control for the amount of code and natural language in the pretraining data, this work assumes that C4 has been fully cleaned of any code data and that ``_cleaned code from Github_’’ does not contain any natural language.

This study only focuses on a relatively small Transformer decoder model of 374M parameters.

---

> ### Author Response · Authors · 2024-11-14
> **Response to reviewer DwKu**
>
> 1. Regarding the code dataset used, we use the code subset of The Pile [1], which is subsequently filtered to include only non-binary files of less than 1MB with common code-related file extensions. We will update the section on dataset construction accordingly to clarify this.
>
> [1]. Gao et al. (2020). The Pile: An 800GB Dataset of Diverse Text for Language Modeling. https://arxiv.org/abs/2101.00027
>
> 2. Regarding fine-tuning analysis, we are able to partially answer this question from a post-hoc analysis of our fine-tuning data. We stored model outputs and accuracy metrics every 1k steps during fine-tuning. We observe that validation and generalization accuracy reach roughly their final values quite early, within the first thousand steps of fine-tuning (i.e., by 1k steps models have ~100% validation accuracy and whatever the final generalization accuracy is). Beyond this point, model performance does not meaningfully change, so any speed-up benefit, if it exists, would need to happen within the first thousand steps of fine-tuning. Given the relatively small training cost (in, eg, FLOPs) of the fine-tuning budget versus the pretraining budget, we suspect that even if such a benefit exists for the higher-code models (or the reverse), the difference will be large enough to have practical importance. We would not be opposed to running an additional set of fine-tuning experiments with a higher temporal resolution on accuracy metrics to study whether or not such a benefit exists within the first thousand steps if the reviewer would believe it to be helpful after reading our post-hoc analysis, and in either case we would be happy to include a discussion of these results in an appendix. (We also point the reviewer to our response to reviewer 9e2w above for a more involved discussion of why we chose this fine-tuning setup and what implications that has)
> 3. Regarding figures 2-6, we felt that since we have relatively few datapoints per condition, reporting the spread of individual points would be clearer and potentially less misleading than showing a measure of variance.
> 4. This is an excellent point, and we will absolutely add a discussion of such models to our related work section.
> 5.  This is a reasonable clarification. Our draft paragraph response, to be included in the section on dataset construction:
>
>
> Though both the code and language datasets we use are intended to be distinct in content type from one another, it is likely that there is a degree of overlap in content type between them. For instance, source code often contains comments and string literals containing natural-language data; in the other direction, though C4 is cleaned using a variety of heuristics, some of which are explicitly designed to exclude code-like data, it is likely that such cleaning attempts are imperfect and therefore there may be a (relatively) small amount of code data in the natural-language data source. We do not perform any additional filtering or cleaning of these data sources. As such, the code data source almost certainly contains some natural language data, just as the natural language source likely contains some un-filtered source code. We will perform an analysis of sample documents from our code data source to get an estimate of how much language data is included in the code source.

---

> > ### Comment · Reviewer_DwKu · 2024-11-14
> > **response about planned changes**
> >
> > Thanks for clarifying some points. I will be happy if you include these clarifications in the paper.
> > No need to run an additional set of fine-tuning experiments with a higher temporal resolution, adding a discussion about this would be enough.

---

### Review · Reviewer_9e2w · 2024-10-30

**Summary Of Contributions:**

The manuscript “How Does Code Pretraining Affect Language Model Task Performance?” presents an empirical study in which the impact of the amount of code in comparison to natural language data in pretraining is evaluated with respect to natural language tasks. The results show a positive impact of code on highly-structured tasks indicating and improvement in compositionality. However, code is detrimental for pure linguistic tasks. On average, over all tasks in BigBench, no average change in performance is seen, i.e., only the variance between tasks increases by being better at some (highly-structured) while worse at others (linguistic, world knowledge).

**Audience:**

Yes

**Broader Impact Concerns:**

There are no broader impact concerns.

**Claims And Evidence:**

Yes

**Requested Changes:**

- Consider running the same experiments with BERT-base to see the impact on size (optional)
- Evaluate the impact of the fixed number of finetuning steps to avoid bad fits (mandatory)

**Strengths And Weaknesses:**

The paper is well-written and easy to follow. The experiment is setup in a clean and rigorous. The perspective through the competitive and additive addition of code is valuable to understand how mixtures with different number of training tokens affect the results.

The only notable limitation is acknowledged by the authors: there are models that are a lot larger than those considered, which arguably have a larger capacity to store more knowledge. Again, arguably, there would still be limits, meaning the general concept of the results should still be visible. Nevertheless, optionally, I would have loved to see the same concept with at least one different model size (BERT-base due to the computational effort?) to see the impact of the size. However, I do not require this for a revision, as the contribution is sound and well-studied

There is one magic number in the parameter choices which I cannot really follow and for which I do not know the impact. In Section 4.2.1, the finetuning is always done for 10k steps. Was this choice validated? Does this lead to over- or underfitting? This needs to be clarified and data on this should be reported, e.g., in an online appendix. In case this shows problems with the fits, a more suitable choice, e.g., something more dynamic with early stopping, should be used.

Minor:
Section 4.1: What context length is used, etc.? Please report hyper parameters of the training to make the manuscript self-contained.
Figure 4 needs subheadings for competitive and additive. Currently, this can only be inferred because additive stops at 50%.

---

> ### Author Response · Authors · 2024-11-14
> **Response to reviewer 9e2w**
>
> We thank reviewer 9e2w for their comments. Regarding the two requested changes:
>
> 1. Regarding running additional experiments with BERT, we see the merit in replicating these experiments at different scales. Given time commitments, we leave it to future work to understand exactly how the effects of code-pretraining are impacted by scale.
> 2. We chose to fine-tune for 10k steps mostly for (a) simplicity of experimental design and (b) to facilitate comparison to previous work which evaluates models fine-tuned on these datasets [see, eg, Petty et al. (2024) 2310.19956]. The concern for over- or under-fitting is a reasonable one. We stored all outputs and accuracy metrics every 1k steps over the course of fine-tuning, and so can do a post-hoc analysis of model performance over time. We observed the following: Over the course of fine-tuning, validation performance on each dataset saturates quite early (that is, if we had employed early-stopping then models would have trained for fewer steps than reported here), and does not diverge. Performance on the generalization set likewise reaches roughly its final value (not necessarily at-ceiling) early on, and does not meaningfully change over the course of fine-tuning. We continued fine-tuning past initial saturation of validation metrics under the hypothesis that models may “grok” performance on the generalization set even if training/validation accuracy had already been saturated, but that appears not to have happened. In any case, there are no qualitative differences that arise in our conclusions if we instead evaluate our models at 1k steps instead of 10k. We will include a discussion of these observations in an updated version of this paper.

---

> > ### Comment · Reviewer_9e2w · 2024-11-14
> > **Feedback on planned changes**
> >
> > Sounds good to me!

---

### Review · Reviewer_RUUZ · 2024-11-02

**Summary Of Contributions:**

The paper proposes an analysis of the role of code in LM's pretraining data. In particular, the extent to which code affects performance in various tasks is studied. The settings studied are as follows: one in which the total volume of data during pretraining is constant (competitive) and another in which the volume of linguistic data is constant. At the same time, varying amounts of code are added (additive). Consequently, the analyses focused on the impact of code on tasks requiring compositional skills and overall performance in many tasks in the BigBench benchmark. The main results suggest that adding code improves model performance in tasks requiring structured output, such as semantic parsing, but may reduce performance in tasks requiring nuanced linguistic or factual knowledge.

**Audience:**

Yes

**Broader Impact Concerns:**

The topic of the paper is absolutely of interest; however, I strongly advise the authors to rethink the formalisation of the problem and improve the recommended points. It would be a shame to leave it as it is, given its potential.

**Claims And Evidence:**

Yes

**Requested Changes:**

After reading the weaknesses and strengths, I have some points to make (I hope they can be of help in improving the paper).

- Study on the effects of introducing code: The increase in code may not uniformly improve performance in all types of tasks. The paper says that it might have a negative impact on tasks that rely heavily on linguistic and factual knowledge, which might limit the practical applications of these pre-trained models depending on the target domain. Have you thought about how to argue this point in a hypothetical real-world application?

- and again. In practical contexts, the balance between code and linguistic data might not be so easily manipulated and the computational costs associated with different data mixtures might affect feasibility and efficiency. Could you provide an estimate of this trade off?


In order to make the paper more robust, I would advise the authors to focus on formalising the problem and schematising it through a more accurate description of the experimental setting (code sharing where available) and finally more robust  and complete ablation studies.

**Strengths And Weaknesses:**

**Strengths:**

Among the strengths are some merits that must necessarily be acknowledged, including the organisation of the paper and the very exhaustive examination methodologies.
In addition, the authors performed very broadly (see BigBench benchmark).
The insights into compositional generalisation and the final conclusions are supported by the experimental setting.

In conclusion, the paper has its merits and I have to be honest, it was very interesting to read although I found some problems with clarity that are written in the weaknesses.

**Weaknesses:**

The paper is very interesting and has its myths but there are some points that were not very smooth and quick to understand. (My interest in the topic prompted re-reading and understanding, but this is not always the case)

I would also strongly advise the authors to share the code (anonymous repo) both to improve transparency and to help a better understanding of the content.

---

> ### Author Response · Authors · 2024-11-14
> **Response to reviewer RUUZ**
>
> We thank reviewer RUUZ for their comments. RUUZ raises two weaknesses of the paper in its current form: (1) that its clarity of exposition could at points be improved, and (2) that the code used to train models is not (yet) public. They request two changes: (3) to discuss the practical application of our findings in domain-specific contexts, and (4) to discuss the difficulties & computational costs of trading off between code and language data. To these points, we respond:
>
> (1). Regarding clarity, we are fully committed to editing and revising the writing of the paper to improve expositional clarity. Action editor TLyS has already suggested additional citations which should be discussed, and we fully agree that these additional sources ought to be included. If reviewer RUUZ has any comments on specific parts of the paper which they found unclear, we would be happy to pay special attention to these areas (as it is sometimes hard to estimate which aspects of a paper will be clear or less so to a new audience).
>
> (2) Regarding open-sourcing the code, we make use of the t5x library for training and evaluation of our models. We use a standard transformer architecture, and will provide in an appendix configurations for recreating the models used.
>
> (3) Regarding practical applications, we suspect that the ideal combination of data sources will depend on the intended domain of use for a model. Though our results show that adding in code mixtures is on average helpful, we do not weight any of the downstream evaluations by perceived importance, which may well be a sensible thing to do for model engineers in non-academic settings. As a concrete example, a chatbot-like model should, according to our results, be trained on more code than is present in popular pretraining corpora like The Pile or Dolma. This recommendation, however, should be caveated with the note that we do not explore how post-training objectives like instruction-tuning or RLHF interact with code-mixture in pretraining; since many LLM models undergo some form of post-training, one would need to study this before being confident that the pretraining code-mixture results translate over directly.
>
> (4) Regarding formalizing trade-offs between language and code data, we suspect that the additive setting (where code, or data of any non-language modality, is added in on top of whatever language data is tractable as a pretraining corpus) will most closely match the ‘expected’ pretraining paradigms of frontier LLMs, where training on as much data as one can is strictly better than training on less*. Assuming this, the most tractable use for our results would be to estimate how much code data would need to be gathered to achieve a target code-mixture given whatever natural-language pretraining corpus has already been collected.
>
> As for efficiency, we suspect that code and language data are largely fungible on a FLOPs basis. Tokens are tokens, regardless of their contents. There may be differences between modalities in terms of, say, long-distance dependencies that could mean that code benefits from a longer (or shorter) context length than language data does, but this probably does not have a meaningful impact on the pretraining setup since (a) its most common to use a single context length for all data and (b) sequence-packing ensures that data is used in a FLOPs-efficient manner even if individual sequences are masked at shorter context lengths.

---

### Comment · Action_Editor_TLyS · 2024-11-05
**Author-reviewer discussion period**

Dear all,

Thanks to all reviewers for getting all their reviews in quickly.  As a reminder to the authors, this is a period to freely discuss various review points and engage with the reviewers.

Best,

AE for Submission 3375

---

### Decision · Action_Editor_TLyS · 2024-12-02

**Recommendation:** Accept with minor revision

**Comment:**

All the reviewers agree that the submission is an interesting and thoughtful study on the impact of including code in pretraining data.  Several reviewers raised comments and suggestions calling for clarifications about the presented work (both about experimental details and context of the presented work relative to previous studies), but all reviewers agree no further experiments are necessary given the authors' rebuttal.

Accepting with minor revisions, could the authors update the submission to reflect the changes agreed upon throughout the discussion?  Just a reminder about this suggestion, as well, which could be described as concurrent work and would also help the paper find its audience:
> There does look to be some overlap between the current manuscript and [1]; thus, I would encourage the authors to contrast their work with [1] (in related work) and explain how the presented experiments are different

While not a required revision, I strongly support Reviewer RUUZ's about sharing the code used to run the described experiments.  Allowing other researchers to easily reproduce and augment the featured experiments will greatly increase the impact of paper in the future.

Thanks,

AE for Submission 3375

**Audience:**

Yes

**Claims And Evidence:**

Yes